# Subsequent Ultrasonographic Non-Visualization of the Ovaries Is Hastened in Women with Only One Ovary Visualized Initially

**DOI:** 10.3390/healthcare10030433

**Published:** 2022-02-25

**Authors:** Edward J. Pavlik, Hannah Fancher, Charles S. Dietrich, John R. van Nagell

**Affiliations:** Department of Obstetrics and Gynecology, Division of Gynecologic Oncology, University of Kentucky Chandler Medical Center-Markey Cancer Center, Lexington, KY 40536-0293, USA; hannah.fancher@uky.edu (H.F.); charles.dietrich@uky.edu (C.S.D.); jrvann2@uky.edu (J.R.v.N.J.)

**Keywords:** transvaginal ultrasound, ovary, age, BMI, menopausal status, visualization, detection, body type, weight

## Abstract

Because the effects of age, menopausal status, weight and body mass index (BMI) on ovarian detectability by transvaginal ultrasound (TVS) have not been established, we determined their contributions to TVS visualization of the ovaries when one or both ovaries are visualized on the first ultrasound exam. A total of 29,877 women that had both ovaries visualized on their first exam were followed over 202,639 prospective TVS exams and 9703 women that had only one ovary visualized on their first exam were followed over 63,702 ultrasonography exams. All images were reviewed by a physician. While non-visualization of both ovaries increased with age in women selected on the basis of the visualization of only one ovary on their first ultrasound exam, one or both ovaries could be visualized in two out of every three women at 80 years of age and more than 50% of women over 80 years of age. At each age, more non-visualizations were associated with women that had only one ovary visualized on their first visit. Having only one ovary visualized on the first exam advanced non-visualizations by an average of ~10 years across all ages and by >20 years in women under 40 years of age. Conclusions: Having only one ovary visualized on an initial ultrasound exam considerably hastens complete non-visualization for this population; however, in these women, ovaries can still be visualized well past menopause, and body habitus is not limiting to TVS ovarian imaging, thus TVS should be considered capable of capturing an ovarian image in two out of every three women at 80 years of age.

## 1. Introduction

The first-line imaging approach utilized by radiologists, gynecologists and gynecologic oncologists for the evaluation of women suspected of having an adnexal mass that could be an ovarian malignancy is transvaginal ultrasonography (TVS) [1]. TVS requires no preparation, is well-tolerated, is free of radiation, can be completed quickly and yields high-quality detailed images of the pelvis [2], including early malignancies of the uterus [3] and ovary [4]. TVS has been reported to be significantly more accurate than bimanual clinical examination in detecting the ovaries, especially since ovaries frequently are not palpable in women over 55 years of age, or in women weighing more than 200 lbs [5,6]. Although useful, safe and in wide clinical application, the ultimate limits of modern ultrasonographic instrumentation to detect human ovaries have not been fully characterized. We have reported on ovarian size determined by TVS in a large screening population [7,8], and recently on how age, body habitus and menopausal status affect detectability when both ovaries are visualized on the first ultrasonography [9]. However, little is known about how the detectability of the ovaries is influenced by age, body habitus and menopausal status when only one ovary is visualized on the first ultrasonography. Importantly, the expectation that the ovarian structure may or may not be viewable by TVS can directly impact patient care, especially if imaging is not performed in circumstances when visualization is possible. For example, if a patient is perceived as too old for the ovaries to be visible by TVS, she may only receive a bimanual exam. Consequently, the detection of adnexal pathology may be missed by the physical exam and treatment delayed. However, with an accurate evidence-based perception of the likelihood of ultrasound being able to visualize the ovaries, the opportunity to receive a timely and safe TVS exam will increase the prospects for a clinical intervention if needed.

The objective of this study was to understand how age, menopausal status, weight and BMI influence TVS visualization of the ovaries in women that have only one ovary present on their initial exam in order to provide accurate expectations for the capability of TVS to successfully visualize ovarian structure.

## 2. Materials and Methods

A total of 49,548 women who enrolled in the University of Kentucky Ovarian Cancer Screening Trial (UKOCST) from January 1987 to November 2021 were evaluated. This prospective cohort trial was approved by the University of Kentucky Institutional Review Board for Human Studies (IRB# 45030) and is registered at ClinicalTrials.gov: NCT04473833. The informed consent was approved institutionally and administered by the study sonographers who were also able to answer questions from participants. Eligibility criteria included (1) all asymptomatic women aged ≥ 50 years, and (2) asymptomatic women aged ≥ 25 years with a documented family history of ovarian cancer in at least one primary or secondary relative. All study participants completed a questionnaire that included medical history, surgical history, menopausal status, hormonal use and family history of cancer, as previously published [3]. Menopause was defined as the absence of menses for 12 months, as described by the National Institute of Aging [10]. Women with a known ovarian tumor or a personal history of ovarian cancer were excluded from the present investigation.

Transvaginal ultrasonography and color Doppler were performed using General Electric Voluson P5, P8 and P10 units with a 4 to 11 mHz vaginal probe on women with an empty bladder. In performing the TVS exam, the transducer was gradually inserted while observing the ultrasound image on a monitor. The urinary bladder was used as a consistent landmark in the pelvis relative to much more variable positions of the uterus and the ovaries for assessing the orientation of the transducer. Three scanning approaches were used to comprehensively assess the pelvis:(a)Side-to-side movements to achieve sagittal imaging;(b)90° rotation to obtain semi-coronal images and angulation of the probe vertically; and(c)Varying the depth of probe insertion to expose different pelvic structures within the field of view. The pelvis was surveyed by slowly sweeping the beam in a sagittal plane from the midline to the lateral pelvic sidewalls, followed by turning the probe 90 degrees into the coronal plane and sweeping the beam from the cervix to the fundus. Landmarks for proving structure consisted of identifying the iliac vessels in the pelvic sidewall and the tubal vessels located posterior and parallel to the fallopian tubes. Pressure was applied to at least three regions of the abdominal surface to achieve bowel repositioning in order to assist visualizations. All images were reviewed by a physician and by at least one of the authors. The study protocol specified that ovaries be measured in three dimensions. Ovarian volume was calculated using the prolate ellipsoid formula (length × width × height × 0.523) [11,12]. Thus, a visualization event was validated by findings that obtained all three measurements. All screening information was entered into a database (MEDLOG Systems, Crystal Bay, NV, USA) on a local network. Women who had a normal screen were scheduled to return in 12 months for a repeat screen. Only women with one or more visible ovaries on their first TVS encounter were utilized in this study and then followed over the course of annual examinations by TVS. Women who underwent abdominal surgery were censored from the present analysis so that surgical interventions did not influence assessments of visualization outcomes.

### Statistical Analyses

Percentages of sonographic findings were determined at the grouping level (i.e., age, weight, BMI, menopausal status). Years to which visualization was shortened due to only one ovary visualized on the first exam were determined as the difference in years between equal percentages of non-visualization on the best-fit curve for both ovaries visualized on the first visit vs. only one ovary visualized on the first visit. Rates, probability ratio, Chi-square test of association with Yates and Pearson *p*-values, Fisher exact probability test, multiple regression analysis and confidence intervals were obtained using Vassarstat based on logistic regression [13].

## 3. Results

A total amount of 29,877 women were identified that had both ovaries visualized on their first TVS exam and were subsequently followed over a course of 202,639 prospective TVS exams. 9703 women were identified that had only one ovary visualized on their first TVS exam and were subsequently followed over a course of 63,697 prospective TVS exams. Demographic characteristics are presented in Table 1.

**Table 1 healthcare-10-00433-t001:** Demographic characteristics of women undergoing TVS. Data represented as mean, median, (range). Subject data are for the first encounter, while encounter data are across all encounters.

	All Subjects (*n* = 29,877)	All Encounters (*n* = 202,639)
Age (y)	55.0, 55 (20–91)	60.1, 60 (20–95)
Weight (kg)	73, 70.3 (38–204)	72.3, 69.4 (36–205)
Height (cm)	163.5, 162.6 (119–198)	163.6, 162.6 (119–198)
BMI	27.3, 26 (13–80)	27, 26 (13–80)
Pre-menopausal	5966 (20.9%)	28,618 (14.3%)
Peri-menopausal	1262 (4.4%)	5820 (2.9%)
Post-menopausal	21,251 (74.6%)	165,390 (82.8%)

Age–non-visualization incidence increased with age (Figure 1A, open red symbols), while visualization of one ovary decreased with age (Figure 1A, solid blue diamonds). Despite selection for women with only one ovary visualized on their first visit, both ovaries were reported about a third of the time subsequently (Figure 1A, solid black symbols). When the women that had neither ovary visualized on the first ultrasound exam were examined (*n* = 7975), a similar fraction was found to have both ovaries reported as visualized subsequently, indicating that there is variability in the identification of ovarian structures over time. Visualization of one or both ovaries decreased with age, but one or both ovaries could be visualized in two out of every three women at 80 years of age and more than 50% of women over 80 years of age (Figure 1A).

**Figure 1 healthcare-10-00433-f001:**
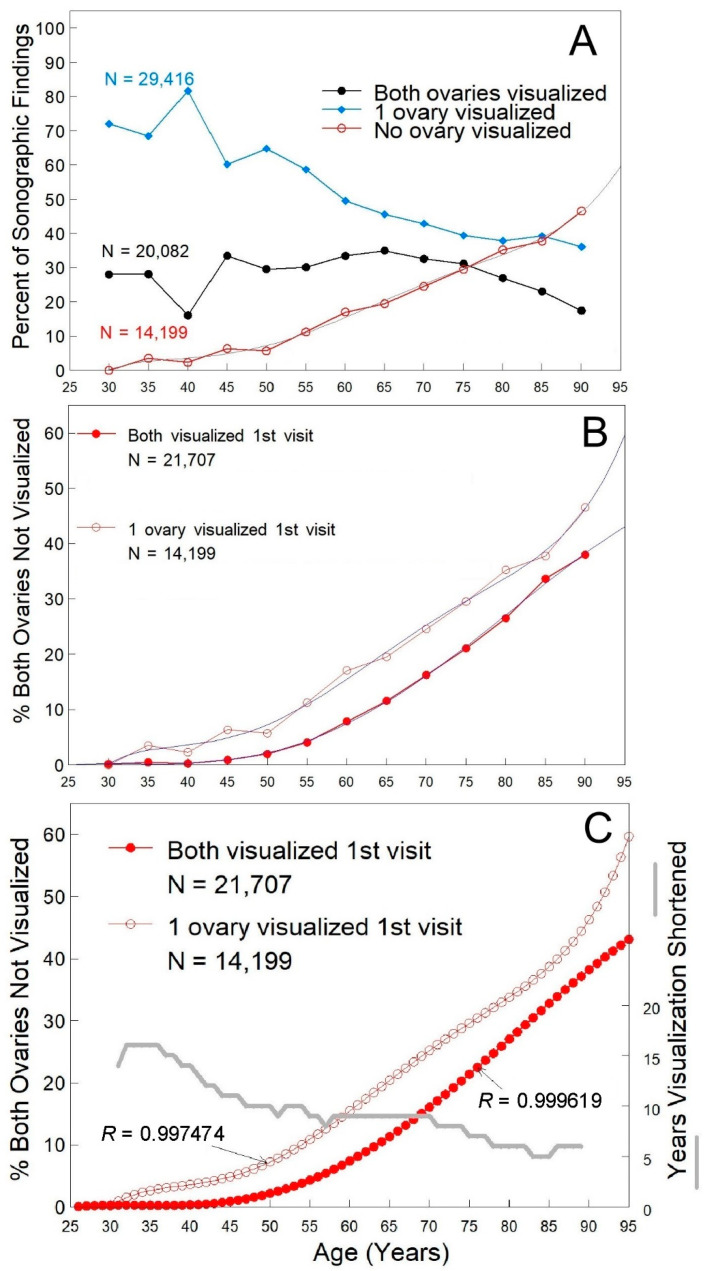
Ovarian visualization relative to age. Panel (**A**). Sonographic findings visualizing both ovaries (black line), one ovary (blue line) or neither ovary (red line) when only one ovary was visualized on the first visit. The line of best fit for visualization of neither ovary is shown in gray (5th degree polynomial, (*R* = 0.997474, y = −324.904195 + 31.873955*X* − 1.205511*X*^2^ − 0.021956*X*^3^ − 0.000191*X*^4^ − 0.00000063935*X*^5^). Panel (**B**). An increase in non-visualization of both ovaries as a function of age was determined when both ovaries were visualized on the first visit (solid red symbols, *R* = 0.999619) or when only one ovary was visualized on the first visit (open red symbols). For when only one ovary was visualized on the first visit, the line of best fit for visualization of neither ovary is shown in blue (5th degree polynomial, (*R* = 0.997474, y = −324.904195 + 31.873955*X* − 1.205511*X*^2^ − 0.021956*X*^3^ − 0.000191*X*^4^ − 0.00000063935*X*^5^), open symbols), while when both ovaries were visualized on the first visit, the line of best fit for visualization of neither ovary is shown in blue (5th degree polynomial, (*R* = 0.999610, y = −33.913696 + 3.453024*X* − 0.126511*X*^2^ − 0.001996*X*^3^ − 0.000012*X*^4^ − 0.00000002428358*X*^5^), solid symbols). Panel (**C**). Shortening of visualization results when only one ovary is observed on the first visit. Best fit curves are shown with the gray curve showing the number of years that non-visualization is advanced when only one ovary is visualized on the first visit. Where *N* = number of TVS observations. Age was self-reported and associated with 29,877 women that received 202,639 TVS encounters when both ovaries were visualized on the first visit and with 9703 women who received 63,697 TVS encounters when only one ovary was visualized on the first visit. Percentages were determined within each age group.

Women that had only one ovary visualized on their first visit were characterized by non-visualization of both ovaries that increased sharply after age 50 (Figure 1A, red line) to a maximum after age 85 of 47%. A crossover point was noted for women in their mid-80s, where non-visualization of both ovaries became greater than any visualization (Figure 1A). The profile for the non-visualization of both ovaries was very well fitted by a 5th degree polynomial (Figure 1A, gray line, *R* = 0.997474) over the range 25–90 years of age. The increase in non-visualizations with age in women that have only one (Figure 1B, open symbols) or both ovaries (Figure 1B, closed symbols) on the first visit revealed that at each age more non-visualizations were associated with women that had only one ovary on their first visit. Using the best-fit curves derived from women that had one (open symbols, Figure 1C) or both ovaries (solid symbols, Figure 1C) visualized on their first visit, the separation in time between the equivalent extent of non-visualizations was determined (Figure 1C, gray line). Non-visualization was advanced by over 20 years in women under 40 years of age (Figure 1C, gray line) and decreased to over 12 years for women 40–70 years old, before decreasing to less than 10 years in women over 70 years of age. Across all ages, non-visualization of both ovaries was advanced by an average of 9.6 years (SEM = 0.41, median = 9 years, 5–16 years range).

Body Habitus–Neither weight (Figure 2A, *p* > 0.2) nor BMI (Figure 2B, *p* > 0.12) was independently associated with ovarian visualization since across all weights and BMIs, non-visualization of both ovaries was ~20–25%. Both ovaries were visualized in 25–30% of women that weighed 180–275 lbs, in 17–23% of women that weighed over 300 lbs (Figure 2A), and in >25% of women with a BMI over 40 (Figure 2B).Menopausal Status–Menopausal status was self-reported. Non-visualization was ~7% in pre-menopausal women and ~25% of postmenopausal women (Figure 2C, *p* < 0.001) that initially had only one ovary visualized.

## 4. Discussion

The visualization of only one ovary on the initial ultrasound exam can hasten non-visualization of the ovaries by a decade or more. In these women, TVS should be considered capable of imaging one or both ovaries in two out of every three women at 80 years of age. Weight, BMI, and menopausal status had little effect on ovarian visualization. The importance of the present report is that it shows even when only one ovary is visualized initially, there is considerable opportunity for ovarian visualization.

### 4.1. Clinical Implications

For women that have only one ovary visualized on their initial ultrasound exam, TVS imaging should be considered viable even for elderly women and not a reason to forego TVS. There is the prospect that eventual non-visualization will occur much sooner in these women, as indicated by the efforts presented here. Nevertheless, age should not be a factor that deters sonographic exploration of the ovaries. Obesity has become more prevalent in American women, increasing from 10% in 1979–80 to 40+% in 2013–14 [14]. This increase has been suggested to present a mounting challenge to obtaining high-quality TVS images of the abdomen [15] because the depth of insonation needed in very large women attenuates the ultrasound beam. TVS reduces the distance between the vaginal transducer and pelvic structures, allowing detailed visualization of the ovaries. TVS lacks ionizing radiation and is not subject to absolute weight/girth restrictions like CT or MRI. TVS provides an opportunity to bypass the abdominal apron of excess skin and fat that characterizes obesity. Glanc et al. have reported that there is a paucity of evidence-based literature on the implications of obesity on TVS [15]. Here, we report that for women having only one ovary visualized on their first ultrasonography, modern TVS instrumentation can achieve equivalent ovarian visualization in both obese and non-obese women.

### 4.2. Research Implications

The results reported here are prospective findings gleaned from long-term data collection in a large ovarian screening study. While it could be possible that newer instrumentation employed over the 30-year course of data collection might influence visualization, there was only a 2.6% increase in the visualization of both ovaries in the most recent 15 years of TVS as compared to the first 15 years. This estimate relies on having sufficient numbers in the groups used to compare effects due to advances in ultrasound technology. Indeed, in the first 5 years of the screening program, 2068 cases were accrued as compared to 24,808 in the most recent 15 years. Visualizations of both ovaries were 30.2% less in the first 5 years than the most recent 15-year experience, indicating that the earliest ultrasound instruments in this study performed less well. This difference decreased to 0–6% with each 10-year continuation of the study. However, the bias represented by these early screens is limited because they account for only 4.2% of the total first screen observations (2068/49,548). Although admission to the study was weighted so that the number of women at least 50 years of age (86%) was greater than women younger than 50 years of age (14%), a substantial number of measurements (*n* = 33,399) were collected for the younger age group. Future investigation should focus on identifying factors correlated with aging that contribute to failures to visualize the ovaries, including bladder and bowel dysfunction.

### 4.3. Strengths and Limitations

The strengths of this paper arise from the large group size of this prospectively studied cohort of asymptomatic women and from criteria for reporting ovarian visualization involving measurements in three planes. By concentrating on women that had only one ovary visualized on their first encounter, this study was able to determine that subsequent non-visualization comes considerably sooner. It is possible that the group studied varies from the symptomatic population seen in the clinic, which may affect generalizability; however, women with abnormal TVS findings are included in this study, as they constitute a visualization event. Women whose ovaries were removed were excluded; however, any bias presented by those that were excluded is small, representing ~5% of the women with only one ovary visualized on the initial TVS exam. There are factors that can confound the findings reported here. First, approximately 85% of the participants had their first TVS exam at age 50–55 years of age, but women younger than this could have had their first exam over a broader age range (age 25–49). Thus, our use of the first TVS event can create variability in participants younger than 50 years that is greater than in women older than 50 years of age. Second, because both ovaries were subsequently visualized after non-visualization of one or both ovaries in about a third of the TVS studies, clinical interpretations should realize that non-visualization is not absolute, but subject to the skills of the sonographer and changes in the physiology of the participant, such as bowel structure and intestinal gas that obstruct visualization. Third, to compensate for variability, measures have been presented as percentages in each category under consideration. Although this has resulted in very high degrees of fit in the curves presented here, it should be kept in mind that the findings in each category should be recognized as idealized. Taken together, the findings reported here are most appropriate at the population level.

## 5. Conclusions

Age has the greatest influence on the visualization of the ovaries. When only one ovary is visualized on the first ultrasound exam, eventual non-visualization can be advanced by a decade or more than when both ovaries are confirmed ultrasonographically on the first exam. Menopause is often perceived as synonymous with aging; however, the ovaries can be visualized well past the onset of menopause. In a majority of women, the ovaries can be visualized for 20–25 years beyond menopause, which occurs on average at age 50 [16,17,18]. Evidence is presented here that the evaluation of the ovarian structure using TVS is possible at almost any age and weight, and even when only one ovary is visualized initially. Thus, older and obese patients remain good candidates for TVS exams, providing a solution to the limitations that occur with physical exams [4,5].

## Figures and Tables

**Figure 2 healthcare-10-00433-f002:**
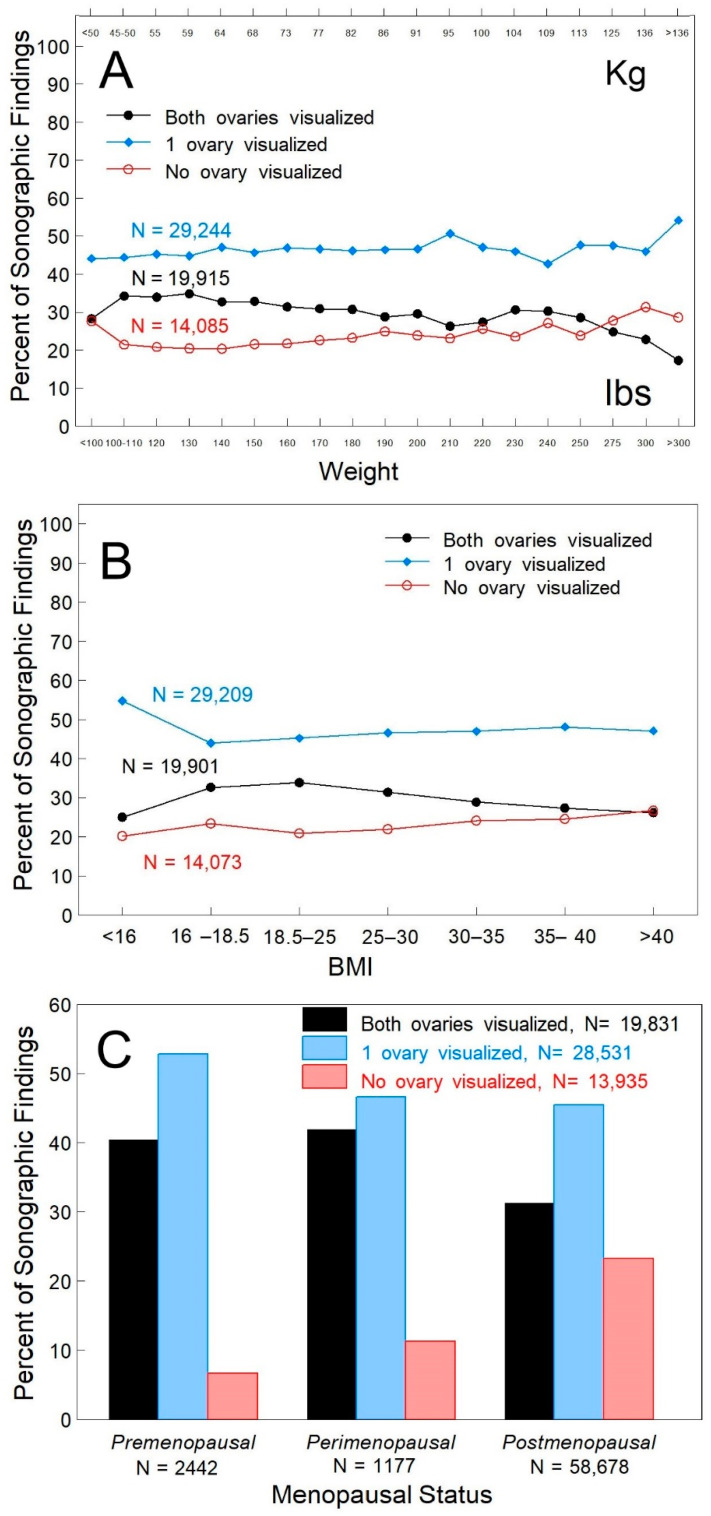
Ovarian visualization relative to weight, BMI and menopausal status when only one ovary is visualized on the first visit. Panel (**A**): Weight was reported in 63,244 TVS encounters. Panel (**B**): Weight and height were reported for the calculation of BMI in 63,183 TVS encounters. Panel (**C**): Menopausal status was self-reported at the time of the TVS exam in 62,297 TVS encounters. Sonographic findings visualizing both ovaries (black bar), one ovary (blue bar) or neither ovary (red bar). Differences in the number of TVS exams reflect unreported events (missing data) resulting in exclusion from analysis. Percentages were determined for each group.

## Data Availability

De-identified data sharing is available after institutional approval by the University of Kentucky and the requestor’s institution/employer through Institutional Review Board approved protocols at each.

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
