# Peer review of "Subsequent Ultrasonographic Non-Visualization of the Ovaries Is Hastened in Women with Only One Ovary Visualized Initially"

_healthcare, 2022, doi:10.3390/healthcare10030433_

Round 1
Reviewer 1 Report
Dear Authors,
I have read and considerably appreciated the manuscript titled Subsequent ultrasonographic non-visualization of the ovaries is hastened in women with only one ovary visualized initially, a research article based on a substantially large sample which relies on solid methodology and rational structuring to convey a highly relevant fundamental message: transvaginal ultrasound is by and large a reliable screening method for elderly and obese patients as well, including two thirds of 80-year-old women.
The implications in terms of screening and diagnostic approaches are adequately described at page 7, although I believe more context ought to be provided in terms of TVS possible applications, particularly in cancer screening.
In light of its narrow focus and competently assembled structure, streamlined, albeit at times overly succint, presentation, I believe the article is worthy of publication and will make for a moderately valuable and original contribution.
One caveat: the references are not compliant with MDPI style requirements, they need to be done over.
In addition to my previous comments, I could add that the authors have failed to take into account transvaginal ultrasound-related confounding factors and how those would apply to and affect outcomes in the cohort they took into account in their study. The discussion should in fact be integrated with an elaboration of relevant confounding factors and how and to what extent the authors have assessed them, given how such dynamics can affect the reliability and the diagnostic accuracy of TVU.
Sincerely,
Author Response
Reviewer 1
The implications in terms of screening and diagnostic approaches are adequately described at page 7, although I believe more context ought to be provided in terms of TVS possible applications, particularly in cancer screening.
We have deliberately avoided expanding the context of this paper to considerations of screening for ovarian cancer using TVS for two reason:
First, we are actively involved in an ongoing ovarian cancer screening protocol that is based on TVS (https://obgyn.med.uky.edu/users/epaul1#profileTab4 https://ukhealthcare.uky.edu/markey-cancer-center/patient-care/cancer-screening-program/ovarian . Second, widely recognized reports in the literature have differed with regard to reporting a survival benefit for the way TVUS was used in several trials.
- van Nagell, J.; Burgess, B.; Miller, R.; Baldwin, L.; DeSimone, C.; Ueland, F.; Huang, B.; Chen, Q.; Kryscio, R.J.; Pavlik, E.J. Survival of Women With Type I and II Epithelial Ovarian Cancer Detected by Ultrasound Screening. Gynecol. 2018, 132, 1091–1100. https://doi.org/10.1097/AOG.0000000000002921. SURVIVAL BENEFIT REPORTED
- van Nagell, J.R.; Miller, R.W.; DeSimone, C.P.; Ueland, J.R.; Podzielinski, I.; Goodrich, S.T.; Elder, J.W.; Huang, B.; Kryscio, R.J.; Pavlik, E.J. Long-term survival of women with epithelial ovarian cancer detected by ultrasonographic screening. Gynecol. 2011, 118, 1212–1221. https://doi.org/10.1097/AOG.0b013e318238d030. SURVIVAL BENEFIT REPORTED
- Buys, S.S.; Partridge, E.; Black, A.; Johnson, C.C.; Lamerato, L.; Isaacs, C.; Reding, D.J.; Greenlee, R.T.; Yokachi, L.A.; Kessel, B.; et al. Effect of screening on ovarian cancer mortality: The Prostate, Lung, Colorectal and Ovarian (PLCO) Cancer Screening Randomized Controlled Trial. JAMA 2011, 305, 2295–2303. https://doi.org/10.1001/jama.2011.766. NO SURVIVAL BENEFIT REPORTED
- Menon, U.; Gentry-Maharaj, A.; Hallett, R.; Ryan, A.; Burnell, M.; Sharma, A.; Lewis, S.; Davies, S.; Philpott, S.; Lopes, A.; et al. Sensitivity and specificity of multimodal and ultrasound screening for ovarian cancer, and stage distribution of detected cancers: Results of the prevalence screen of the UK Collaborative Trial of Ovarian Cancer Screening (UKCTOCS). Lancet Oncol. 2009, 10, 327–340. https://doi.org/10.1016/S1470-204570026-9. NO SURVIVAL BENEFIT REPORTED
- Menon, U.; Gentry-Maharaj, A.; Burnell, M.; Singh, N.; Ryan, A.; Karpinskyj, C.; Carlino, G.; Taylor, J.; Massingham, S.K.; Raikou, M.; et al. Ovarian cancer population screening and mortality after long-term follow-up in the UK Collaborative Trial of Ovarian Cancer Screening (UKCTOCS): A randomised controlled trial. Lancet 2021, 397, 2182–2193, https://doi.org/10.1016/S0140-673600731-5. NO SURVIVAL BENEFIT REPORTED
- Kobayashi, H.; Yamada, Y.; Sado, T.; Sakata, M.; Yoshida, S.; Kawaguchi, R.; Kanayama, S.; Shigetomi, H.; Haruta, S.; Tsuji, Y.; et al. A randomized study of screening for ovarian cancer: A multicenter study in Japan. J.Gynecol. Cancer 2008, 18, 414–420. https://doi.org/10.1111/j.1525-1438.2007.01035.x. NO SURVIVAL BENEFIT REPORTED
In light of its narrow focus and competently assembled structure, streamlined, albeit at times overly succint, presentation, I believe the article is worthy of publication and will make for a moderately valuable and original contribution.
One caveat: the references are not compliant with MDPI style requirements, they need to be done over.
Changes to the references have been made to make them compliant with MDPI guidelines
In addition to my previous comments, I could add that the authors have failed to take into account transvaginal ultrasound-related confounding factors and how those would apply to and affect outcomes in the cohort they took into account in their study. The discussion should in fact be integrated with an elaboration of relevant confounding factors and how and to what extent the authors have assessed them, given how such dynamics can affect the reliability and the diagnostic accuracy of TVU.
We have added considerations related to confounding factors in lines 242-255:
“There are factors that can confound the findings reported here. First, approximately 85% of the participants had their first TVS exam at age 50-55 years of age, but women younger than this could have had their first exam over a broader age range (age 25-49). Thus, our use of the first TVS event can create variability in participants younger than 50 years that is greater than in women older than 50 years of age. Second, because both ovaries were subsequently visualized after non-visualization of one or both ovaries in about a third of the TVS studies, clinical interpretations should realize that non-visualization is not absolute, but subject to the skills of the sonographer and changes in the physiology of the participant like bowel structure and intestinal gas that obstruct visualization. Third, to compensate for variability, measures have been presented as percentages in each category under consideration. Although this has resulted in very high degrees of fit in the curves presented here, it should be kept in mind that findings in each category should be recognized as idealized. Taken together the findings reported here are most appropriate at the population level.”
Reviewer 2 Report
The authors examined the effect of patients’ characteristics on the detection rate of ovaries by transvaginal ultrasonography. This study is interesting and worth to be published. I have some suggestions to improve the manuscript.
I recommend that the manuscript be reviewed by a professional native English speaker to rectify the errors in grammar, punctuation, word choice, and sentence construction. This would improve the flow of information and ensure that the document reads as though written by a native English speaker.
The paragraph on clinical implications is a run-on paragraph that is excessively lengthy. Long-winding sentences tend to confuse readers and may lead to misinterpretation. Short sentences are preferred for improved clarity and readability. Therefore, I suggest revising this segment using shorter sentences.
Since the study period is wide (1987 to 2021), the authors need to determine the effect of the year of inspection (e.g. 1987-1998 versus 1999-2010 versus 2011-2021) on the detection rate of ovaries.
Figure 1
Please do not include the formula in the panel.
Table 1
]] should be ].
Author Response
Reviewer 2
The authors examined the effect of patients’ characteristics on the detection rate of ovaries by transvaginal ultrasonography. This study is interesting and worth to be published. I have some suggestions to improve the manuscript.
I recommend that the manuscript be reviewed by a professional native English speaker to rectify the errors in grammar, punctuation, word choice, and sentence construction. This would improve the flow of information and ensure that the document reads as though written by a native English speaker.
We have shared the manuscript with several fluent English speakers and incorporated changes to improve the English writing with these changes made at:
Lines 49-50, 52, 62-64,109,111-112, 120, 122, 142-144, 147-148,150, 153, 157-158,164,178-181, 194,197-199, 201-210, 213-216,223-224, 227, 229-238, 247-248, 254-268, 270-281. Please note that line numbers are for when track changes are “on” or visible.
The paragraph on clinical implications is a run-on paragraph that is excessively lengthy. Long-winding sentences tend to confuse readers and may lead to misinterpretation. Short sentences are preferred for improved clarity and readability. Therefore, I suggest revising this segment using shorter sentences.
The section on clinical implications has been edited to shorten sentence structure and avoid run-on sentences over the lines 201-224. Please note that line numbers are for when track changes are “on” or visible.
“Clinical Implications
For women that have only one ovary visualized on their initial ultrasound exam, TVS imaging should be considered viable even for elderly women and not a reason to forego TVS. There is the prospect that eventual non-visualization will occur much sooner in these women as indicated by the efforts presented here. Nevertheless, age should not be a factor that deters sonographic exploration of the ovaries.
Obesity has become more prevalent in American women, increasing from 10% in 1979-80 to 40+% in 2013-14. This increase has been suggested to present a mounting challenge to obtaining high quality TVS images of the abdomen because the depth of insonation needed in very large women attenuates the ultrasound beam. TVS reduces the distance between the vaginal transducer and pelvic structures allowing detailed visualization of the ovaries. TVS lacks ionizing radiation and is not subject to absolute weight/girth restrictions like CT or MRI. TVS provides an opportunity to bypass the abdominal apron of excessive skin and fat that characterizes obesity. Glanc et al have reported that there is a paucity of evidence-based literature on the implications of obesity on TVS. Here we report that for women having only one ovary visualized on their first ultrasonography, modern TVS instrumentation is capable of achieving equivalent ovarian visualization in both obese and non-obese women.“
Since the study period is wide (1987 to 2021), the authors need to determine the effect of the year of inspection (e.g. 1987-1998 versus 1999-2010 versus 2011-2021) on the detection rate of ovaries.
We have introduced modifications at lines 230-238 that characterize the effect of year of inspection, focusing on the first five years of the screening program and each 10 years of continuation. Please note that line numbers are for when track changes are “on” or visible.
“This estimate relies on having sufficient numbers in the groups used to compare effects due to advances in ultrasound technology. Indeed, in the first 5 years of the screening program, 2068 cases accrued as compared to 24,808 in the most recent 15 years. Visualizations of both ovaries were 30.2% less in the first 5 years than the most recent 15 year experience, indicating that the earliest ultrasound instruments in this study performed less well. This difference decreased to 0-6% with each 10 year continuation of the study. However, the bias represented by these early screens is limited because they account for only 4.2% of the total first screen observations (2068/49,548). Although admission to the study was weighted so that the number of women at least 50 years of age (86%) was greater than women younger than 50 years of age (14%), a substantial number of measurements (n=33,399) were collected for the younger age group.”
Figure 1
Please do not include the formula in the panel.
The formula/equation has been removed. See lines143-144,147- 148, 150
Please note that line numbers are for when track changes are “on” or visible.
Table 1
]] should be ].
Correction has been made within Table 1
Round 2
Reviewer 2 Report
The reviewer thinks the authors revised the manuscript well.